# Pharmaconutrition in the Clinical Management of COVID-19: A Lack of Evidence-Based Research But Clues to Personalized Prescription

**DOI:** 10.3390/jpm10040145

**Published:** 2020-09-25

**Authors:** Heitor O. Santos, Grant M. Tinsley, Guilherme A. R. da Silva, Allain A. Bueno

**Affiliations:** 1School of Medicine, Federal University of Uberlandia (UFU), Uberlandia 38408-100, Brazil; 2Department of Kinesiology and Sport Management, Texas Tech University, Lubbock, TX 79409, USA; grant.tinsley@ttu.edu; 3Hospital Universitário Gaffrée e Guinle, Federal University of the State of Rio de Janeiro (UNIRIO), Rio de Janeiro 21941-901, Brazil; drguialmeida@gmail.com; 4College of Health, Life and Environmental Sciences, University of Worcester, Worcester WR2 6AJ, UK; a.bueno@worc.ac.uk

**Keywords:** severe acute respiratory syndrome coronavirus 2, SARS-CoV-2, coronavirus disease 2019, COVID-19, nutrition, supplementation

## Abstract

A scientific interest has emerged to identify pharmaceutical and nutritional strategies in the clinical management of coronavirus disease 2019 (COVID-19). The purpose of this narrative review is to critically assess and discuss pharmaconutrition strategies that, secondary to accepted treatment methods, could be candidates in the current context of COVID-19. Oral medicinal doses of vitamin C (1–3 g/d) and zinc (80 mg/d elemental zinc) could be promising at the first signs and symptoms of COVID-19 as well as for general colds. In critical care situations requiring parenteral nutrition, vitamin C (3–10 g/d) and glutamine (0.3–0.5 g/kg/d) administration could be considered, whereas vitamin D3 administration (100,000 IU administered intramuscularly as a one-time dose) could possess benefits for patients with severe deficiency. Considering the presence of n-3 polyunsaturated fatty acids and arginine in immune-enhancing diets, their co-administration may also occur in clinical conditions where these formulations are recommended. However, despite the use of the aforementioned strategies in prior contexts, there is currently no evidence of the utility of any nutritional strategies in the management of SARS-CoV-2 infection and COVID-19. Nevertheless, ongoing and future clinical research is imperative to determine if any pharmaconutrition strategies can halt the progression of COVID-19.

## 1. Introduction

With the emergence of severe acute respiratory syndrome coronavirus 2 (SARS-CoV-2)/coronavirus disease 2019 (COVID-19) worldwide, alongside substantial concern regarding the mortality rates in 2020 [1], extensive biomedical research has been devoted to mitigate this pandemic. In addition to the basic protective measures against the new coronavirus recommended by the World Health Organization (WHO) [2] (e.g., washing hands and social distancing), there is a fervent scientific interest in evaluating effective pharmacotherapy treatment of COVID-19 [3,4]. These endeavors are not limited to pharmaceutical agents, as several recent reports have considered the potential role of nutraceutical agents and proposed dosing regimens for vitamins, minerals, amino acids, and herbal medicines in this context [5,6,7]. However, the question remains: to what extent, if any, are nutraceutical agents and pharmaconutrition strategies useful against COVID-19 when applied in tandem with recommended clinical management?

While the term *nutraceuticals* is broadly defined and relatively vague [8,9], this category of substances can include dietary supplements, in general, herbal medicines and “functional foods”. In contrast, pharmaconutrition is considered as a clinical nutrition therapy based on nutrients commonly provided in medical care via oral, enteral (i.e., administration into gastrointestinal tract), or parenteral (i.e., administration into the bloodstream) routes [10,11]. Given the more rigorous implementation of pharmaconutrition, and its importance for outpatient and inpatient treatments, we performed a literature review focusing on the use of well-known nutrients possessing pharmacological effects, particularly in critical care environments such as the intensive care unit (ICU), which could possibly be useful in the present COVID-19 pandemic due to their previously established clinical outcomes and safety record [12,13,14].

## 2. Methods

We employed a literature search using the Medline/PubMed database to identify preferentially meta-analyses of randomized clinical trials (RCTs) that investigated the administration of nutrients with pharmacological effects used in some clinical settings, particularly in critical care environments such as the intensive care unit (ICU). In addition, we considered original studies in order to complement meta-analyses. Evidence was reviewed regarding the administration of vitamin C (ascorbic acid), vitamin D, zinc, omega-3 (n-3) polyunsaturated fatty acids (PUFA), glutamine, and arginine, as these are nutritional elements with physiological importance for the immune system and may be found within immune-enhancing diets (i.e., a formula containing arginine, n-3 PUFA, glutamine, nucleotides, and structured lipids); additionally, some of these nutraceutical components possess in vitro antiviral effects [15,16]. We primarily discuss recent meta-analyses to inform and encourage better clinical decision-making of practitioners and to provide insights for future COVID-19 RCTs.

## 3. Vitamin C

Apart from the pivotal role of endogenous antioxidants, antioxidant supplements have drawn attention against pathogens in the immune system [17,18,19]. Vitamin C is a potent antioxidant that scavenges oxygen free radicals and restores other cellular antioxidants; as such, this nutrient is suggested to improve viral-induced oxidative injury [20].

Indeed, there is a close link between vitamin C and the immune system as this nutrient is considered an essential factor in the function of phagocytes, transformation of T lymphocytes, and production of interferon [21]. Accordingly, vitamin C is highly concentrated in white blood cells such as lymphocytes and macrophages [20]. Due to the increased inflammatory response and metabolic demand during infectious diseases as well as their associations with low circulating vitamin C concentrations, vitamin C administration may be beneficial in combatting a number of viral infections mainly by increasing the production of α/β interferons and downregulating the production of pro-inflammatory cytokines [20,22].

Regarding clinical effects, in a meta-analysis of RCTs that examined ≥0.2 g/d of oral vitamin C, there was no reduction in the incidence of colds, but in adults, the duration of colds was decreased by 8% (95% confidence interval (CI): 3% to 12%) [23]. Apart from this incipient outcome, the impact of administering vitamin C under intensive care deserves attention in order to ascertain whether or not it meaningfully influences primary endpoints. In this regard, one meta-analysis demonstrated that vitamin C administration reduced general ICU length of stay (LOS) by 8% on average [24]. In trials employing oral vitamin C administration, a total of six reports using 1–3 g/d (weighted mean 2 g/d) decreased the general ICU LOS by 8.6% (95% CI: 3.0% to 14.0%). In addition, three trials in which patients required mechanical ventilation for over 24 h indicated that vitamin C administration shortened the duration of mechanical ventilation by 18.2% (95% CI: 7.7% to 27%). However, it is worth mentioning that in this meta-analysis [24], 18 controlled trials were selected regardless of placebo treatment, and 13 of these studies were carried out on patients undergoing elective cardiac surgery.

Vitamin C therapy in ICU conditions generally consists of high doses administered for short periods (i.e., days or weeks), whereas a putative prophylaxis against viral infections may require extended periods (i.e., several months). Accordingly, high-dose vitamin C administration should only receive consideration as a potential complementary therapy in the context of critical care, while a nutritious diet including green vegetables, citrus fruits (e.g., oranges, lemons, grapefruit, Persian lime), and other foods containing this nutrient is sufficient to maintain adequate vitamin C status in the general population. Importantly, it should be emphasized that the pharmacokinetic characteristics differ between oral and intravenous administration of vitamin C. Accordingly, intravenous administration can yield 30-to 70-fold higher blood concentrations of vitamin C than the maximum tolerated oral dose [25]. Ultimately, clinical trials should determine whether hospitalized patients infected with SARS-CoV-2 show low plasma concentrations of vitamin C in order to inform the potential usefulness of administering this micronutrient.

## 4. Zinc

Zinc supplementation is recommended for the treatment of various ailments as a means of modulating anti-inflammatory and antioxidant pathways in several systems as well as regulating the immune system by T cell-mediated functions [26,27]. In general, a large body of evidence shows that up to one year of pharmacological zinc doses (220 mg/d to 660 mg/d in chelated form, corresponding to approximately 50 mg to 150 mg of elemental zinc) seems to be a safe strategy in various clinical conditions, patient populations (i.e., from children to the elderly), and for patients with complicated diseases such as liver and kidney disease or diabetes mellitus [26,28,29,30,31]. The viability of supplementing zinc in elderly patients may be particularly salient due to the elevated mortality of aged patients with COVID-19 when compared with young and middle-aged subjects [32]. Moreover, preclinical data suggests that intracellular zinc inhibits the replication of SARS-CoV [33].

Zinc’s best-documented effects against viral illness are likely those pertaining to the common cold. In a meta-analysis of placebo-controlled trials [34], acute medicinal provision of zinc reduced the duration of the common cold by approximately three days. In that analysis, trials that employed zinc acetate lozenges in an elemental dose > 75 mg/d were included; however, only three studies were eligible due to exclusion of studies employing low doses of zinc. Importantly, participants from these studies were instructed to administer one lozenge (9–13 mg of elemental zinc per lozenge) every 2–3 h while awake, thereby achieving a dose of ~80 mg/d elemental zinc for one to two weeks. Lozenge use began immediately after the first common cold symptoms [35,36,37]. While zinc lozenges may be a viable option that could be offered to patients, the high frequency of daily administration necessary to achieve a medicinal dosage could be impractical; conversely, administration is recommended for only 1–2 weeks in an attempt to improve an acute clinical condition.

In another meta-analysis using medicinal doses of zinc, the mean common cold duration was 33% (95% CI: 21% to 45%) shorter for those receiving zinc supplementation [38]. Interestingly, Hemilä separately analyzed two zinc salts; three trials resulted in 40% shorter cold duration when using acetate zinc lozenges, and four trials resulted in 28% shorter cold duration when using zinc gluconate. However, the apparent 12% difference between the zinc forms was not definitive (95% CI: −12 to +36%). Regarding the elemental zinc doses, five trials [35,36,37,39,40] used 80–92 mg/d and two trials [41,42] used 192–207 mg/d, which reduced the common cold duration by 33% and 35%, (LOS respectively, without statistical difference between doses. Hence, the use of zinc doses over 100 mg/d are not necessary in this context.

## 5. Vitamin D

Vitamin D, a micronutrient possessing some known hormonal actions, has proposed anti-viral effects; however, the interplay between vitamin D and viral infections is yet to be fully elucidated [43,44]. Nonetheless, it appears that vitamin D promotes antiviral activity in airway epithelial cells during infection [45,46]. Additionally, low vitamin D status is associated with higher rates of upper and lower respiratory tract infections. From a broader perspective, severe 25-hydroxyvitamin D (25(OH)D) deficiency is associated with many chronic diseases [47,48]. While the challenge of defining whether low 25(OH)D status is the cause or consequence of specific chronic diseases remains, severe deficiency (<10 ng/mL), but not deficiency (10~20 ng/mL) or insufficiency (20~30 ng/mL), at point of admission is independently associated with increased risk of mortality in patients with sepsis [49]. At the same time, worsening severity of 25(OH)D deficiency is associated with increased LOS and mortality rate in general surgery patients admitted to the surgical ICU as well as higher ICU treatment cost [43].

Upon administration of vitamin D in the VITdAL-ICU trial [50], lower hospital mortality was noted in critically ill patients with severe 25(OH)D deficiency who received high-dose vitamin D3 when compared to placebo. Notwithstanding this result, it is critical to note that the study design involved a 540,000 IU vitamin D3 dose via oral or nasogastric tube followed by monthly maintenance doses of 90,000 IU for five months, producing a total follow-up time of six months, thus being potentially inapplicable to the acute care of patients infected with COVID-19. In one meta-analysis [51] encompassing six RCTs with 695 critically ill patients, no differences in infection rate, ventilation days, LOS in the ICU or hospital, or mortality were found with vitamin D administration. Even >300,000 IU of vitamin D3 daily, irrespectively of route (orally, via naso/oro-gastric tube, or intramuscularly) did not improve mortality. Overall, when viewed collectively, there is insufficient evidence to affirm that vitamin D administration improves clinical outcomes in critically ill patients.

Importantly, a recent study [52] examining 449 patients with COVID-19 from UK Biobank detected an initial association between low 25(OH)D levels and COVID-19 (OR = 0.99; 95% CI: 0.99–0.999), but not after adjustment for confounders (OR = 1.00; 95% CI: 0.998–1.01). This finding ultimately does not support a link between vitamin D concentrations and COVID-19. Additionally, while race/ethnicity was associated with COVID-19 univariably (Blacks vs. Whites OR = 5.32, 95% CI: 3.68–7.70; South Asians vs. Whites OR = 2.65, 95% CI: 1.65–4.25), adjustment for 25(OH)D levels did not meaningfully alter the magnitude of the associations.

Given the paucity of scientific support for administration of vitamin D in the context of COVID-19, it is reasonable to screen 25(OH)D levels in patients with immune disturbances so that, in the case of deficiency, recommended practices can be followed. The Endocrine Society recommendations [44] state that the following doses should be employed in the event of deficiency: (1) 50,000 IU of vitamin D2 or D3 once a week for 8-wk program; (2) the equivalent of 6000 IU/d of vitamin D2 or D3, with maintenance therapy of 1500–2000 IU/d; or (3) 100,000 IU of vitamin D every four months.

## 6. Omega-3 Polyunsaturated Fatty Acids

Biochemically, PUFA appears to inactivate human-infecting enveloped viruses, and there is an association of higher intake of PUFA with reduced risk of pneumonia [53,54]. The n-3 PUFA are known to influence aspects of innate and adaptive immunity through a variety of mechanisms including beneficial effects on the cell membrane and roles in cell signaling [55]. As such, several investigations have sought to establish whether n-3 PUFA influences clinical outcomes in critical care settings.

Overall results from the meta-analysis of Koekkoek et al. [56] did not substantiate significant effects of enteral fish oil supplementation on 28-day ICU or hospital mortality in critically ill patients. However, ICU LOS and the ventilation duration were significantly reduced in patients who received fish oil supplementation, and there was a significant reduction in 28-day mortality in patients with acute respiratory distress syndrome (ARDS). Notwithstanding these benefits, the authors reported a low methodological quality of the studies performed on ARDS, which was seemingly caused by heterogeneity and some differences in the controlled diets such as the addition of other immunomodulatory substances (e.g., antioxidants, arginine, and glutamine).

Another meta-analysis examining critically ill patients with ARDS was also performed [57]. Compared to the placebo, n-3 PUFA added to enteral immunomodulatory diets improved early (3–4 d) and late (7–8 d) arterial partial pressure of oxygen (PaO_2_)/fractional inspired oxygen (FiO_2_) ratio, a marker used to quantify the severity of the ARDS, but mortality, hospital LOS, and infectious complications did not change. In addition, the authors reported trends for the beneficial effects in those who received n-3 PUFA with respect to reduced ICU LOS (*p* = 0.08) and length of mechanical ventilation (*p* = 0.06). Ultimately, the authors [57] concluded that administering n-3 PUFAs appears to be viable strategy in ARDS. In another meta-analysis of septic patients [58], n-3 PUFA administration failed to reduce mortality and the LOS of hospital and intensive care, but the duration of mechanical ventilation was shortened (weighted mean difference = −3.82; 95% CI: −4.61 to −3.04).

At a minimum, n-3 PUFA administration could be considered in COVID-19 critical care situations based on its safety, common occurrence in immune-enhancing diets, and the ability to promote adequate n-3 PUFA status in the body. However, quantification of circulating levels of n-3 PUFA is a potential dilemma, as they are not usually measured in routine clinical practice. Correspondingly, only six of the 24 studies of the meta-analysis of Koekkoek et al. reported plasma levels of n-3 PUFA [56]. However, taking into account an insufficient intake of n-3 PUFA in many countries, particularly in those in which the general population habitually consumes fish at a low frequency, it may be admissible to employ n-3 PUFA as a complement to adequate nutrition of a critically ill patient regardless of blood levels. Overall, the n-3 PUFA dose for patients with ARDS in the meta-analysis of Koekkoek et al. was based on 5.3 g/L eicosapentaenoic acid (EPA) via enteral nutrition, with only one study using n-3 PUFA soft gels (720 mg) [56]. Therefore, the observed effects and proposed use are seemingly directed at EPA.

Unfortunately, the majority of clinical studies that have investigated the effects of n-3 PUFA administration did not test this nutrient alone, as n-3 PUFA was included within immune-enhancing diets or other nutritional strategies [56,57]. Importantly, Stapleton et al. examined the effect of n-3 PUFA administration, without other nutritional intervention, in patients with acute lung injury [59]. In a randomized placebo-controlled design, patients who received enteral fish oil administration (9.75 g EPA and 6.75 g docosahexanoic acid [DHA] daily) for up to 14 days did not present significant change in bronchoalveolar lavage fluid interleukin (IL)-8 from baseline to day 4 or day 8 when compared to the placebo (saline infusion). Likewise, organ failure score, ventilator-free days, ICU-free days, and 60-day mortality did not differ between the groups.

## 7. Arginine

Arginine is a non-essential amino acid in the normal physiological state (i.e., it is sufficiently synthesized in the human body to meet the needs for growth and health [60]). However, arginine becomes conditionally essential during critical periods such as, for example, when burn patients are receiving total parenteral nutrition [61,62]. Mechanistically, arginine and its related pathways play a pivotal role in the pathophysiology of respiratory illnesses (e.g., asthma), mainly via the enzymatic inter-relationships between arginase and nitric oxide synthase (NOS) [63]. Within this biochemical reaction, there is competition between both enzymes for arginine as a substrate, from which nitric oxide (NO), a small gaseous signaling molecule, is produced by isoforms of NOS while arginase decreases NO biosynthesis [64]. Patients with pulmonary arterial hypertension show increased arginase level and decreased NO synthesis when compared to healthy controls [65]. In addition, patients with cystic fibrosis also present dysregulation in the arginine metabolic pathway, as demonstrated by a study [66] in which reduced systemic bioavailability of arginine was detected in this population compared to healthy non-smokers.

In the aforementioned clinical conditions, circulating arginine levels may be considered as a surrogate disease biomarker; however, this does not necessarily imply that administration of arginine is a viable treatment. For instance, in patients with cystic fibrosis, systemic arginine concentrations were normalized after two weeks of antibiotic treatment used for pulmonary disease exacerbation [66]. Concerning arginine supplementation in pulmonary disorders, some studies [67,68] have demonstrated metabolic and clinical benefits in patients with asthma and pulmonary hypertension such as increases in plasma arginine levels and reduction of estimated pulmonary artery systolic pressure. Nevertheless, these are preliminary findings based on a limited number of individuals. Cumulatively, effects of arginine administration on lung function are potentially plausible and could be investigated in patients with SARS-CoV-2, as severe cases of this disease are associated with pneumonia and ARDS by virtue of worsening inflammatory-induced lung injury [69]. At a minimum, this amino acid is important for stimulating immune function via its impact on lymphocytes and macrophages, and its administration is proposed to attenuate clinical infections and shorten hospital LOS [61]. While many dietary sources of arginine are present in the food supply, higher doses of 15 to 30 g/d of supplemental arginine are commonly used for critically ill patients [61]. Therefore, in a similar way to n-3 PUFA, arginine could be administered within immune-enhancing diets when deemed appropriate.

Despite indications that supplemental arginine administration is safe in the aforementioned dosages [70], caution should be exercised due to potential drug-supplement interactions. For example, risk of hypotension in the event of concurrent administration of arginine alongside blood pressure medication could result in cardiopulmonary instability in patients with concurrent cardiac dysfunction and/or pulmonary hypertension.

## 8. Glutamine

Glutamine is a conditionally essential amino acid that serves as a critical fuel source for cells of the immune system [71]. Additionally, the high demand for glutamine can cause a reduction in free glutamine concentrations during periods of pronounced physiological stress including major infection [72]. Glutamine supplementation is a strategy that may be useful in clinical settings due to its immunomodulatory effects [73]. The “cytokine storm” observed in patients with COVID-19 refers to an early response of pro-inflammatory cytokines, specifically tumor necrosis factor-α (TNF-α), IL-6, and IL-1β [74]. Interestingly, glutamine administration could be a candidate for attenuating this cytokine storm as it may reduce the release of pro-inflammatory cytokines such as TNF-α, IL-6, and IL-8, while increasing the concentrations of IL-10, an anti-inflammatory cytokine [75,76].

In a meta-analysis of RCTs addressing critically ill adult patients (i.e., major surgery, trauma, infection, or organ failure), glutamine dipeptide administration via parenteral route (0.3–0.5 g/kg/d) significantly reduced infectious complications (relative risk (RR) = 0.70, 95% CI: 0.60, 0.83), ICU LOS (−1.61 days, 95% CI: −3.17, −0.05), hospital LOS (−2.30 days, 95% CI: −4.14, −0.45), mechanical ventilation duration (−1.56 days, 95% CI: −2.88, −0.24), and mortality rate (RR = 0.55, 95% CI: 0.32, 0.94), but had no effect on ICU mortality [77]. Importantly, this proposed glutamine regimen is in accordance with clinical guidelines [78,79] and occurred as a complement to isoenergetic and isonitrogenous nutrition therapy. Alternatively, enteral glutamine supplementation does not generally confer significant clinical benefits in critically ill patients, as demonstrated in a meta-analysis that included only RCTs employing this route of administration [80]. Although there was a significant benefit in the subset of patients with burns, the pathophysiology of this condition is not comparable with the aggravated presentation observed in COVID-19.

It is not advisable to extrapolate glutamine supplementation as a means of improving immune function of the general population, since the primary indication for glutamine administration relates to critical medical conditions [81]. Correspondingly, a recent meta-analysis demonstrated that glutamine supplementation had no effect on athletes’ immune systems [82]. Ultimately, a balanced diet is sufficient to maintain adequate glutamine status in healthy individuals, and supplementation should not be viewed as a beneficial preventative strategy. In critical COVID-19 cases, clinicians and researchers could consider examination of glutamine concentrations in order to further inform whether administration of this amino acid could hold a potential benefit for improving clinical outcomes.

## 9. Pharmaconutrition in Patients with Diabetes

Recently, in a cohort of 7337 COVID-19 patients, those diagnosed with diabetes demonstrated an increased need for medical interventions and higher mortality risk, with well-controlled blood glucose levels being correlated with better outcomes [83]. When considering the pharmaconutrition strategies discussed in this review, therapeutic doses of glutamine may reduce blood glucose concentrations of outpatients with diabetes and mitigate the hyperglycemia of critically ill diabetic patients under total parenteral nutrition [84]. Moreover, medicinal dosages of zinc and vitamin C appear to reduce blood glucose levels in patients with diabetes, as supported by the results of recent meta-analyses [29,85,86]. However, these effects have been observed in non-COVID-19 situations.

Regarding mechanisms of action, glutamine is an efficacious glucagon-like peptide (GLP)-1 secretagogue, and when administered with a meal, increases both GLP-1 and insulin secretion, resulting in reduced postprandial blood glucose in subjects with type 2 diabetes [84]. As such, it was demonstrated that glutamine administration mitigated the hyperglycemia of critically ill diabetic patients on total parenteral nutrition [84], a route of administration frequently associated with high levels of blood glucose [87]. The effects, if any, of vitamin C and zinc on blood glucose have not been clearly elucidated; however, their antioxidant potential could potentially be beneficial as a means of decreasing reactive oxygen species, which are produced intensely in patients with diabetes [88,89].

## 10. Strengths and Limitations

In contrast with other recent reviews [5,6,7], we intentionally did not focus on herbal medicines since their use is uncommon in intensive care. Conversely, vitamins, minerals, and amino acids are essential for the body, are utilized in medical care settings, and may be administrated beyond the oral route. Most importantly, the ongoing pharmaconutrition trials for the treatment of COVID-19 are focusing on vitamins and minerals (Table 1). The focus on nutrients with established utility in clinical nutrition as well as those currently being examined in preliminary COVID-19 trials is a strength of the present review as we attempted to avoid misconceptions and unsubstantiated recommendation of substances without applicability in critical situations. However, it is also recognized that herbal medicines are a pertinent tool as an adjuvant to medical therapies in specific outpatient scenarios (e.g., type 2 diabetes mellitus [90], dyslipidemia [91,92] etc.)

Despite a number of meta-analyses [24,34,38,51,56,57,58,77,80] addressing the use of pharmaconutrition to combat illness, most are limited due to the studies’ heterogeneity (Table 2). Unfortunately, heterogeneity is a common concern in meta-analyses [93]. Thus, while awareness of this and other limitations is essential, meta-analytical techniques are still a fundamental component for critically appraising the existing evidence and informing decision-making that, in turn, covers the wide diversity observed in clinical practice.

## 11. Summary and Recommendations

In light of the lack of scientific basis underpinning new nutrition strategies for the management of COVID-19 and its related consequences, the focus of the present work was to examine nutrients with immunological properties commonly used in the treatment of hospitalized patients, as summarized in Figure 1 through a diagram of tentative proposals. Prior to widespread implementation of pharmaconutrition strategies, it is crucial to verify the effects of the tentatively proposed therapies on recognized outcomes such as ICU and hospital mortality and inherited outcomes (e.g., ICU and hospital LOS, ventilation duration and infectious complications).

To the best of our knowledge and considering the current scenario, oral vitamin C (1–3 g/d) and zinc (80 mg/d elemental zinc) could be considered at the first signs and symptoms of COVID-19; parenteral vitamin C (3–10 g/d) and glutamine (0.3–0.5 g/kg/d) could be considered in critical care situations; vitamin D3 administration (100,000 IU as a one-time dose) could be appropriate for patients with severe deficiency, and co-administration of n-3 PUFA and arginine could be appropriately recommended in clinical conditions where immune-enhancing diets are recommended. Although blood levels of zinc do not necessarily indicate the intracellular zinc status [25], evaluating its concentrations as well as vitamin C and 25(OH)D may be important before administration, as these biomarkers are used in medical care settings by way of comparison with other nutrients and could influence subsequent clinical decision-making.

Overall, the appraised meta-analyses considered all pharmaconutrition strategies discussed in this review to be generally safe for administration to ill patients [23,33,37,51,56,57,58,77,80]. However, pharmaconutrition strategies should be employed under the guidance or supervision of qualified medical personnel, particularly given that the doses of minerals and vitamins considered may exceed the Tolerable Upper Intake Level (e.g., >40 mg/d zinc, >2 g/d vitamin C, >2000 IU/d vitamin D) [94], while the parenteral route may be required for critical care settings. Prudence must be exercised so as not to culminate in unnecessary polypharmacy, which may affect the quality of life of patients and cause undesired reactions due to drug-nutrient interactions [95,96].

Certainly, the administration of these nutrients should not be viewed as a sole strategy for the treatment of COVID-19 as it is critical to follow expert recommendations, ranging from basic protective measures endorsed by the WHO, to established medical guidelines in intensive care settings. The identification and treatment of COVID-19 patients therefore involve a comprehensive team of health professionals (physician, nurse, dietitian, pharmacist, physical therapist, etc.). Finally, pharmaconutrition treatment is suggested as a possible acute support strategy, but not for prophylactic or long-term use.

## 12. Conclusions

To date, there is no direct evidence regarding the efficacy of particular nutraceutical strategies in the management of SARS-CoV-2 infection and COVID-19. However, given the relevance of the current global crisis and the knowledge that adequate nutrition is required for all members of society, decision-making should be based on the best existing scientific knowledge in this scenario. As discussed herein, oral medicinal doses of vitamin C and zinc may potentially hold promise at the first signs and symptoms of COVID-19 as well as for general colds. Concerning critical care situations linked to COVID-19, not only is vitamin C administration plausible, but also glutamine when parenteral nutrition is recommended, while vitamin D3 administration could be useful for patients with severe deficiency. As n-3 PUFA and arginine are commonly present in immune-enhancing diets, their co-administration may occur in clinical conditions when these formulations are recommended. Collectively, while the clinical approaches are relatively well documented in prior contexts, these nutritional candidates only afford possible adjuvant effects to complement the recommended medical treatment of COVID-19. Further clinical research including the ongoing trials highlighted in the present review, should explore these nutritional strategies in the context of COVID-19 to establish new options for combating the devastating effects of this pandemic.

## Figures and Tables

**Figure 1 jpm-10-00145-f001:**
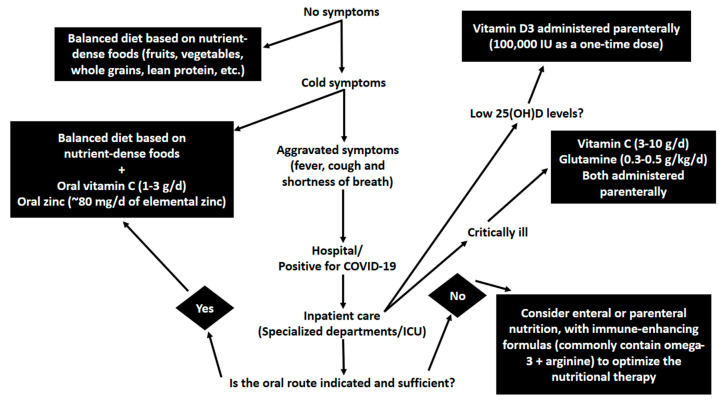
Possible adjunct pharmaconutrition in the management of COVID-19. All pharmaconutrition strategies displayed in this diagram should be considered in conjunction with established medical care guidelines. Additionally, such proposed roles are based on non-COVID-19 research and reflect the authors’ interpretation of potentially relevant literature. Interventions should ideally be further assessed in clinical trial and cautiously administered under close medical care.

**Table 1 jpm-10-00145-t001:** Ongoing pharmaconutrition-related trials, as registered on https://clinicaltrials.gov/. Search carried out on 4 August 2020.

Interventions	Conditions	Estimated Enrollment (n)	Phase	Country	Clinical Trial ID
Vitamin C 2-h infusion daily (for 6 d), escalating dose (0.3 g/kg, 0.6 g/kg, 0.9 g/kg).	Hospitalized patients with COVID-19	66	Recruiting	USA	NCT04363216
50 mg/kg vitamin C infusion given every 6 h for 4 d (16 total doses)	COVID-19 and hypoxia	20	Recruiting	USA	NCT04357782
10 g of IV vitamin C in addition to conventional therapy	Hospitalized patients with COVID-19 pneumonia	500	Recruiting	Italy	NCT0432351
Inpatients: IV vitamin C (Sodium Ascorbate) 50 mg/kg every 6 h on d 1 followed by 100 mg/kg every 6 h (4×/d; 400 mg/kg/d) for 7 d (average 28g/d). Outpatients: 200 mg/kg ×1 IV vitamin C, then 1 g PO 3×/d for 7 d. Plus active comparator: 400 mg PO 2×/d Hydroxychloroquine, followed by 200mg PO 2×/d for 6 d. 500 mg/d PO azithromycin followed by 250 mg/d PO for 4 d. 30 mg/d PO (elemental dose) zinc citrate. 5000 IU/d PO vitamin D3 for 14 d. 500 mcg/d PO vitamin B12 for 14 d.	COVID-19	200	Not yet recruiting	Australia	NCT04395768
IV vitamin C: 50 mg/kg every 6 h for 96 h (16 doses).	Hospitalized patients with COVID-19	800	Not yet recruiting	Canada	NCT04401150
Methylene blue, vitamin C, N-acetyl cysteine	COVID-19	20	Recruiting	Iran	NCT04370288
Drug: hydroxychloroquine Dietary supplements: vitamin C, vitamin D and zinc. Use as a prophylaxis treatment for COVID-19	COVID-19	600	Not yet recruiting	USA	NCT04335084
100 mg/kg intravenous vitamin C infusion every 8 h for up to 72 h	COVID-19 Lung Injury, Acute	200	Not yet recruiting	USA	NCT04344184
Experimental: oral loading dose of 800 mg followed by once weekly oral hydroxychloroquine 400 mg for 3 mo. Active comparator: oral vitamin C 1 g/d for 3 mo.	COVID-19	1212	Not yet recruiting	USA	NCT04347889
12 g vitamin C 2×/d for 7 d with infused pump speed of 12 mL/h.	COVID-19 pneumonia, ventilator-associated	140	Recruiting	China	NCT04264533
Comparator: ascorbic acid 500 mg orally daily for 3, then 250 mg orally daily for 11 d Experimental: hydrochloroquine 400 mg orally daily for 3 days, then 200 mg orally daily for an additional 11 days	COVID-19	2000	Not yet recruiting	USA	NCT04328961
Quintuple therapy for 24 weeks Drugs: hydroxychloroquine and azithromycin Dietary supplements: vitamin C, vitamin D, zinc	COVID-19	600	Not yet recruiting	USA	NCT04334512
Plaquenil 200 mg tablet. Proflaxis using hydroxychloroquine + Vitamin C, D and zinc (Not specified dosage)	COVID-19	80	Recruiting	Turkey	NCT04326725
Daily oral nutrition supplement with: 1.1 g EPA, 450 mg DHA, 950 mg GLA, 2840 IU vitamin A as 1.2 mg β-carotene, 205 mg vitamin C, 75 IU vitamin E, 18 µg selenium, and 5.7 mg zinc. Taken 3 h after breakfast.	COVID-19	30	Not yet recruiting	Saudi Arabia	NCT04323228
Vitamin C: 50 mg/kg every 6 h for 96 h.	COVID-19, Sepsis, ICU	800	Recruiting	Canada	NCT03680274
8000 mg of ascorbic acid divided into 2–3 doses/d with food. 50 mg of zinc gluconate to be taken daily at bedtime. Combined and single treatment.	COVID-19	520	Enrolling by invitation	USA	NCT04342728
Vitamin C 3 g/d, 400 mg tiamine, selenium, omega-3500 mg/d, Vit A, Vit D, Azithromycine, Ceftriaxone, Kaletra 2×/d for 10 d.	COVID-19	80	Recruiting	Iran	NCT04360980
Single dose of 25,000 UI of vitamin D supplement in addition to prescription of NSAIDs, ACE2 inhibitor, ARB or thiazolidinediones, according to clinician criteria. Vitamin D supplementation will be taken in the morning together with a toast with olive oil.	COVID-19	200	Not yet recruiting	Spain	NCT04334005
Experimental: 400,000 IU vitamin D3 in a single oral dose. Active Comparator: 50,000 IU in vitamin D supplementation a single oral dose	COVID-19	260	Not yet recruiting	France	NCT04344041

ACE2 inhibitor: angiotensin-converting enzyme 2 inhibitor; ARB: angiotensin II receptor blocker; DHA: docosahexaenoic acid; EPA: eicosapentaenoic acid; ICU: intensive care unit; IV: intravenous therapy; GLA: gamma-linolenic acid; NSAIDs: nonsteroidal anti-inflammatory drugs.

**Table 2 jpm-10-00145-t002:** Summary of main findings of the meta-analyses appraised in the current review. Collectively, those studies addressed pharmaconutrition strategies through randomized clinical trials with medical outcomes that show potential in the management of COVID-19.

Author, Year (Reference)	Supplement, Route	Dose	Duration	Condition	Trials (n)	Patients (n)	I^2^	Main Results
Hemilä and Chalker, 2013 [18]	Vitamin C, oral	≥0.2 g/d	According to mean of cold episodes	General community	29	11,306	38%	In adults, the duration of colds was reduced by 8% and in children by 14%
Wang et al., 2019 [19]	Vitamin C, intravenous	450 mg/d to 66 mg/kg/h	12 h to 28 days	Critically ill	12	1210	0%	Reduced the duration of vasopressor support and mechanical ventilation. 3–10 g vitamin C resulted in lower overall mortality rates
Hemilä et al., 2016 [28]	Zinc, oral	80–92 mg/d elemental zinc dose, acetate lozenges	According to cold episodes	Common cold	3	199	61%	36% (3 days) estimates for the reduction of common cold duration
Hemilä, 2017 [32]	Zinc, oral	80–207 mg/d zinc acetate or gluconate	According to cold episodes	Common cold	7	575	77%	Common cold duration was 33% shorter for the zinc groups
Langlois et al., 2018 [45]	Vitamin D, oro-enteral or parenteral	50,000–540,000 IU	Single dose to 5 days	Critically ill	6	695	0%	No differences in ICU and hospital LOS, infection rate and ventilation day
Tao et al., 2016 [52]	Omega-3, parenteral or enteral	According to commercially available omega-3 fatty acids enriched nutrition (Omegaven^®^, Oxepa^®^, Lipolus^®^)	Varied across different trials	Sepsis	11	11,808	40% for mechanical ventilation and 0% for mortality	Reduced mechanical ventilation duration but not mortality
Koekkoek et al., 2019 [50]	Omega-3, enteral	0.68–16.5 g/L	Varied across different trials	Critically ill	24	3574	2% for overall mortality and 0% for mortality in patients with ARDS	Enteral fish oil supplementation did not change 28-d mortality in general, but reduced mortality in patients with ARDS
Langlois et al., 2019 [51]	Omega-3, enteral or parenteral	According to commercially available omega-3 fatty acids enriched nutrition (Omegaven^®^, Oxepa^®^, Ultimate Omega^®^). Overall, 1.3 g/d to 10.2 g/formula	Varied across different trials	Critically ill patients with ARDS	20	1280	69% for PaO_2_-to-FiO_2_ ratio	Improved early and late PaO_2_-to-FiO_2_ ratio
van Zanten et al., 2015 [74]	Glutamine, enteral	0.27 to 0.5 g/kg/d	Varied across different trials	Critically ill	11	1079	52% for LOS	There was no reduction of hospital mortality, infectious complications, or stay in the ICU, but there was a significant reduction in LOS (~5 days)
Stehle et al. 2017, [71]	Glutamine dipeptide, parenteral	0.4–0.5 g/kg/d alanylglutamine (alanine-glutamine)	3–21 days	Critically ill	16	842	0% for hospital mortality, 31% for infectious complication, 0% for LOS	Reduced hospital mortality, infectious complication rates, and hospital LOS

Overall, when the value of the I^2^ statistic is greater than 50%, it can be regarded as heterogeneity [93]. Legend: ARDS, acute respiratory distress syndrome; ICU, intensive care unit; LOS, length of stay; PaO_2_-to-FiO_2_ ratio, arterial partial pressure of oxygen/fractional inspired oxygen ratio.

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
