# Peer review of "Pharmaconutrition in the Clinical Management of COVID-19: A Lack of Evidence-Based Research But Clues to Personalized Prescription"

_jpm, 2020, doi:10.3390/jpm10040145_

Round 1
Reviewer 1 Report
Overall, this is a well-written and documented paper on the topic. I would only have some minor suggestions.
I would start each paragraph with an overview of the known roles of each molecule reviewed. For example, vitamin D is a compound with effect1, effect2, effect3 (e.g. antineoplastic, antiobesity, antioxidant etc). Please see the following papers:
https://www.hindawi.com/journals/omcl/2014/158135/
https://www.tandfonline.com/doi/abs/10.1080/01635581.2020.1797127
https://www.sciencedirect.com/science/article/abs/pii/S104366181500290X
https://www.sciencedirect.com/science/article/pii/S1018364720300203
Also, regarding pharmaconutrition, I would also raise the point of polypharmacy in diabetes. Please see:
https://www.mdpi.com/1010-660X/55/8/436
Overall, this is an excellent paper.
Author Response
Dear reviewer, thank you very much for your kind words and helpful comments! We have cited all the references suggested. Meta-analyses on zinc and vitamin D were used to support their topics. The references on antioxidants were relevant in order to expand mechanisms of actions. Lastly, in topic 10, we have created a new sentence based on the polypharmacy paper.
Reviewer 2 Report
The manuscript “Pharmaconutrition in the clinical management of COVID-19: a lack of evidence-based research but clues to personalized prescription” explore the potential of pharmaconutrition strategies for COVID-19 patients. This is a well-written manuscript presenting relatively comprehensive information regarding the pros and cons of different nutrients.
Minor comments:
- Line 64, please define the acronym “RCT” if this is the first time used in the manuscript;
- Line 65, please change “attenuated” to another word;
- Line 87, please change “COVID-19” to "SARS-CoV-2. Infected with viruses, not diseases.
- Line 157-158, the statement is contradictory with evidence;
- Line 160-161, please rephrase this sentence;
Author Response
Dear reviewer, the authors express sincere gratitude for your considerations. Indeed, we have incorporated your suggestions and mended the text accordingly.
Reviewer 3 Report
This is a narrative review on Pharmaconutrition in the clinical management of COVID-19
Unfortunately, despite the efforts of the authors there is nothing new.
I suggest doing a systematic review on the same topic editing the paper for english-language usage.
Author Response
Dear reviewer, unfortunately, we cannot mend the paper as suggested within the very tight timeframe. There is no time to carry out a systematic review, but we have added a topic of methods in an attempt to meet your expectations.
We respectfully believe that a systematic review is not appropriate for this topic because several agents have been considered. Systematic reviews may be unactionable in the current scenario insofar as COVID-19 is a transient disease. So much so that there are very few RCT on Covid-19 specifically, and at the moment a meta-analysis of RCT would be underpowered. However, we have applied a convenient method for narrative review and therefore the paper has gained a better arrangement.
Regarding the English-language usage, thank you for pointing this out. We have double-checked the paper for grammar and structure and incorporated changes wherever we felt it was necessary.